# An Analysis of Urban Block Initiatives Influencing Energy Consumption and Solar Energy Absorption

Ngakan Ketut Acwin Dwijendra [1,*], Untung Rahardja [2], Narukullapati Bharath Kumar [3,*], Indrajit Patra [4], Musaddak Maher Abdul Zahra [5], Yulia Finogenova [6], John William Grimaldo Guerrero [7], Samar Emad Izzat [8] and Taif Alawsi [9]

1. Department of Architecture, Faculty of Engineering, Udayana University, Bali 80361, Indonesia
2. Faculty of Science and Technology, University of Raharja, Banten 15117, Indonesia
3. Department of Electrical and Electronics Engineering, University of Vignan's Foundation for Science, Technology and Research, Guntur 522213, India
4. NIT Durgapur, Durgapur 713209, India
5. Computer Techniques Engineering Department, Al-Mustaqbal University College, Hillah 51001, Iraq
6. Department of State and Municipal Finance, Plekhanov Russian University of Economics, Moscow 117997, Russia
7. Department of Energy, Universidad de la Costa, Barranquilla 080001, Colombia
8. Department of State and Municipal Finance, Al-Nisour University College, Baghdad 10001, Iraq
9. Scientific Research Center, Al-Ayen University, Thi-Qar 64001, Iraq
* Correspondence: ngakanketut669@gmail.com (N.K.A.D.); narukullapatibharathkumar@outlook.com (N.B.K.)

**Abstract:** Population growth and urbanization cause developing-country cities to create energy-intensive buildings. Building energy efficiency can be improved through active and passive solar design to reduce energy consumption, increase equipment efficiency, and utilize renewable energy, converting renewable energy into thermal energy or electricity. In this study, passive architecture was evaluated for both urban block and building energy usage. When reliable information and analysis of signs and parameters impacting energy consumption are available, designers and architects can evaluate and passively design a building with higher precision and an accurate picture of its energy consumption in the early stages of the design process. This article compares the location of Baku's building mass to six climate-related scenarios. Three methodologies are used to determine how much solar energy the models utilize and the difference between annual heating and cooling energy consumption. The structure's rotation has little effect on the energy utilized in most forms. Only east-west linear designs employ 6 to 4 kWh/m$^2$ of area and are common. Most important is the building's increased energy consumption, which can take several forms. The building's westward rotation may be its most important feature. Any westward revolution requires more energy. Building collections together offers many benefits, including the attention designers and investors provide to all places. Having an integrated collection and a sense of community affects inhabitants' later connections. Dictionary and encyclopedia entries include typology discoveries. These findings will inform future research and investigations. An architect must know a variety of qualities and organizations to define and segregate the environment because architecture relies heavily on the environment. This research involves analyzing the current situation to gain knowledge for future estimations. The present will determine the future.

**Keywords:** solar energy; maximum energy efficiency; energy consumption reduction

## 1. Introduction

Buildings consume around 48% of global energy use. Most high-energy buildings are being created in urban areas due to population increase, increasing urbanization, and rising living standards. A sustainable growth path based on renewable energy sources and energy side effects requires the eradication of energy resources and climate change [1–3]. Flexible

building design or solar design can help reduce energy usage while achieving modern living standards [4,5]. Buildings play a vital role in providing adequate protection with thermal and visual comfort to their residents [6,7]. Urban architects and designers have been researching the relationship between energy use and the structure of neighborhood units since the nineteenth century [8,9]. The interaction of buildings with their environment is a typical challenge for architects, urban designers, and meteorologists. Meteorologists are interested in how urban development affects climate change [10,11]. At the building scale, architects investigate comfort and energy consumption. According to Figure 1, architects and urban planners determined that focusing solely on buildings without taking into account the surrounding buildings is of little benefit, and hence studies on the thermal behavior of buildings at the level of urban blocks and groups of buildings should be done [12,13]. When a building is positioned in a row, its thermal behavior differs from when it is put alone and without surrounding structures [14]. The effect of the geometry of the blocks on energy consumption is an essential feature of block design in urban development. Meanwhile, numerous studies on neighborhood units and the influence of their shape on energy usage have been conducted [15,16]. A multigeneration system consisting of a double-flash geothermal power plant, solar power tower, organic Rankine cycle, alkaline electrolyzer, and a single-stage absorption chiller was presented. In this research, the solar power tower was used to improve the performance of the double-flash geothermal power plant. Combining such a concentrated solar power system with a double-flash geothermal power plant was not previously addressed. Moreover, to increase the whole system's efficiency, the waste heat from the double-flash geothermal power plant and the solar power tower in the proposed system were used. The proposed system is capable of producing power, hydrogen, heating, and cooling [17].

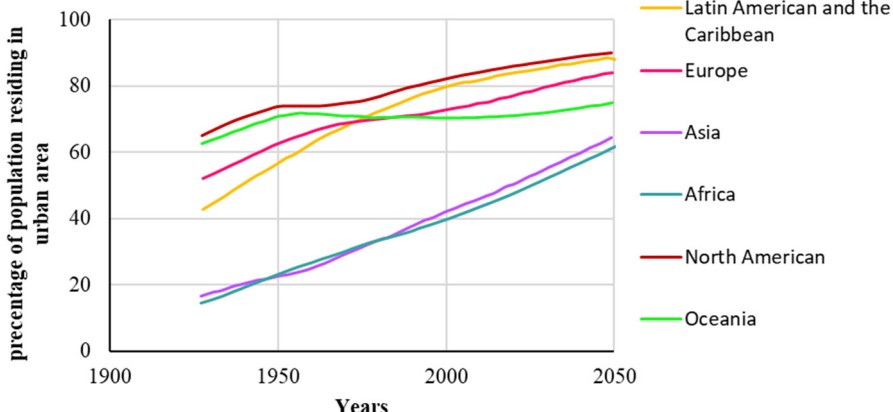

**Figure 1.** The proportion of the world's population living in cities is growing.

With the increased consumption of fossil fuels and the resulting increase in greenhouse gases, as well as the detrimental impacts on the life and safety of living organisms, the use of renewable resources has been incorporated in country macro-policies, including the use of renewable energy [18]. Solar energy holds a distinct place in the world. To accomplish this, the ideal sites to deploy solar panels must be selected [19]. Determining the amount of energy received in any portion of the territory is also essential in this regard. Important elements influence the distribution of solar radiation energy in a region, the most important of which is the format of each region [20]. Elevation differences, plate reasons (slope of their orientation), and the prevention of impacts surrounding an energy point are all elements that contribute to significant local gradients [21,22]. Many studies have identified slope and direction as the primary factors in determining the radiant energy of the sun. The reason for this is the relative ease with which solar radiation may be estimated; in addition to the topography and geometry of the earth, additional factors such as air passage and the relative location of the sun influence this distribution [23,24]. Due to the environmental, economic, and social benefits that can come from a global infrastructure, the building of

a contemporary teaching infrastructure has become a common global objective in recent years. The efficient use of power on the demand side, in particular, plays an essential role in improving energy sustainability and conservation for home users [25]. Recently developed ICT (information and communications technology) improvements include advanced metering infrastructure (AMI), intelligent sensor technologies, telecommunications, two-way, smart home appliances, and home energy storage systems (HESS), among others. As a result, this expanding trend provides the technical foundation and infrastructure for smart homes equipped with home energy management systems [26]. Demand management systems in the smart grids of various home appliances based on user preferences in houses via human–device interaction smart monitors save energy and boost efficiency [27]. Because of growing worries about global energy security and the widespread use of environmentally friendly energy production technologies such as wind turbines, solar panels, and electric vehicles (PEVs), artificial intelligence can be used [28] to increase house energy conversion and operating efficiency. The implementation of these modifications will result in substantial shifts in modern energy management systems, from centralized infrastructure to stand-alone systems powered by renewable energy sources [29]. For buildings and houses in society, there is a necessity to modernize building systems and materials. The quality of the building, the speed of construction, the lifespan of buildings, the reduction of energy consumption, and the reduction of costs have all been prioritized more than before [30]. On the other hand, with rising energy consumption in the construction sector and diminishing energy resources, optimizing energy usage is becoming increasingly important [31].

The amount of sunlight is one of the most essential climate components and is critical for human thermal comfort both indoors and outdoors. Access to solar radiation was one of the most essential aspects in the construction of ancient towns. Indigenous architecture can be seen all over the world [32]. On cold days, the solar architecture allows access to sunshine within the building and on the walkway. Ignoring a building's solar rights and the open spaces around it may result in a loss of thermal comfort [33]. Solar energy can be used not only as static energy for orientation and daylight, but also dynamically for the building's power and hot water usage; different characteristics determine the quantity of access to sunlight [34,35], such as a building's direction and shape, as well as its density and composition of construction. As a result, architects and designers must consider both the shape of the structure and its surroundings early in the design process [36]. As a result, this article examines the literature on design variables and indicators that influence the quantity of solar energy received in order to optimize energy usage in the building. It is recommended when zoning urban blocks to approach the construction of numerous buildings together, rather than as a single building [37]. The outcome is divided areas and detached dwellings in which residents and builders employ all of the power and skills provided by the rules to reap the benefits of better light and more uniqueness in an urban environment. This will result in a type of construction in which the interior and architecture of the building are just as significant as how it looks [38,39].

Building collections together has a number of advantages, one of which is the emphasis that designers and investors bring to the entirety of the collection, in addition to the care that is paid to all spaces. One further advantage that has an effect on the residents' subsequent connections is the achievement of an integrated collection coupled with a sense of common community for the residents. The findings of foundational studies such as typology can be found in dictionaries and encyclopedias. These findings will be of great use in subsequent research and studies since they will provide background information. An architect needs to be knowledgeable about a wide variety of characteristics and organizations so that the environment can be defined and segregated in accordance with those characteristics and organizations. This is because the environment is such an important component of architecture, owing to the fact that data pertaining to the species is one of the most fundamental components of the investigation. This research is based on determining the existing circumstance in order to obtain knowledge, which will then be used to produce

projections regarding the future situation. In other words, the current situation will be used to determine what will happen in the future.

Section 2 introduces the study area and describes how the design of urban blocks affects the amount of sunshine; the theoretical side of the models and model formulation are described in detail. In Section 3, the primary findings of this study are reported, including those on urban density, building direction, and building and street design. Section 4 outlines the conclusions that were drawn from this study.

## 2. Materials and Methods

In this section, we will look at how to estimate the total amount of solar radiation on the earth's surface and how the design of urban blocks affects the amount of sunshine available in order to make the best use of solar energy.

### 2.1. Calculating the Total Amount of Solar Radiation Emitted by the Earth's Surface

The renowned Angstrom equation was the first proposed relationship for determining total sun radiation. Angstrom found a linear link between total solar radiation and sundials. Later, scholars from around the world tweaked the Angstrom connection. Equations (1) and (2) demonstrate:

$$\frac{H}{H_0} + a + b\frac{n}{N} \tag{1}$$

$$\frac{T.S.R}{R_0} = a + b\frac{n}{N} \tag{2}$$

$H$ = $T.S.R$ is the amount of daily radiant energy per unit of horizontal surface in this relation, and Ho is the amount of daily radiant energy per unit of surface in the horizontal surface above the atmosphere, which is $H_0 = R_0$. a and b are the dimensionless coefficients of daily sunshine and the number of possible hours of sunshine per day or day length. Researchers from several countries have obtained air coefficients. The World Food Organization (FAO) requires sunlight energy to assess its evapotranspiration potential. As a result, in arid locations where radiometric data cannot be collected, the values $a = 0.35$ and $b = 0.55$ or an altitude-related quadratic connection can be used [40]. The values of a and b act as a function of latitude, altitude, and the daily hours of sunlight to day length ratio.

### 2.2. The Impact of Urban Block form on the Amount of Sunshine Available

The design of community units based on solar energy availability is dependent on three major factors: the building density of neighborhood units, the building direction, and the roadway design. Key elements such as building shape, building density on site, and site design all play a role in a building's energy use. The impact of a building and the buildings surrounding it on access to solar energy is the highest [41,42]. Several studies about green buildings have been conducted to investigate the effect of urban block shape on the amount of sunshine available. It was explored whether or not there was a connection between the morphology of the city and the amount of daylight that each building received. In addition, the urban pattern of solar residential blocks was investigated and evaluated, and the most common architectural styles of urban blocks were contrasted (Figure 2) [43]. After analyzing the amount of direct sunshine received by various urban levels at a latitude of 25 degrees, it was concluded that solar residential blocks had the highest amount of radiant energy absorption on the roof and facade of the building, among other shapes. New forms are being made for cities with the lowest energy use and the most solar energy input (Figure 3) [44]. The amount of access to sunlight is affected by numerous factors relating to the shape of a building and the buildings surrounding it. These include urban density, building facade direction, and building-to-street relationships (ratio of building height to street width). Access to sunlight in the urban setting and buildings is influenced by urban design factors (street width and urban density) as well as building design elements (building facade and building shape) [45]. Architects and urban planners must understand the potential to receive radiant energy. The design of a building and street's contours is a

critical topic in urban bioclimatic design. Another key aspect influencing solar radiation penetration into buildings is the orientation of the building's facade. A building's facade is a common part of architecture and urban design, serving as the interface between architectural and urban scales [46].

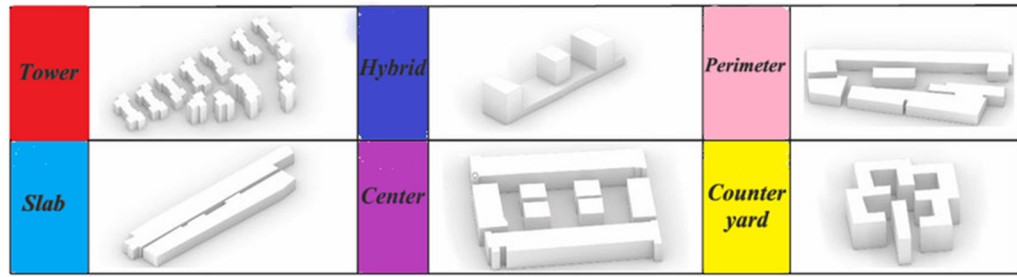

**Figure 2.** There are many different kinds of urban blocks.

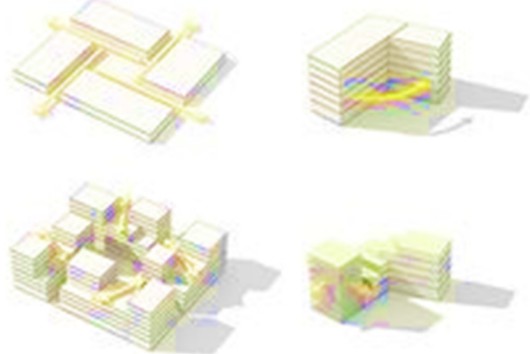

**Figure 3.** New patterns are emerging in urban development that are designed to make efficient use of the sun's energy.

During the phase of the project devoted to plan design, seven potential designs for the climate of the northern hemisphere were developed, and it was concluded that the depth ratio has a significant influence on non-focal types of solar radiation. The U and L forms, for example, benefit from less sunlight than other configurations since the wings cast a shadow on those other surfaces (Figure 4).

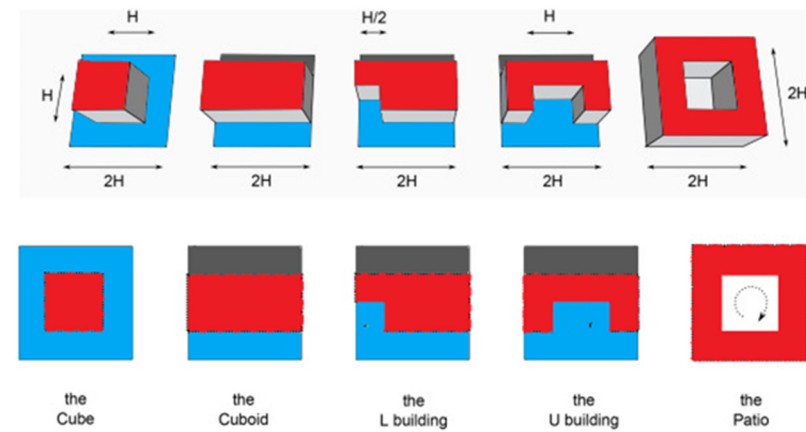

**Figure 4.** The simplest form that each floor plan type can take.

## 3. Results and Discussion

The primary findings of this study are reported in this section, including those on urban density, building direction, and building and street design.

### 3.1. The Density of Cities

One of the most important components in the equation is urban density, which has a considerable impact on access to sunlight. Increased urban structure density limits the quantity of daylight that may reach nearby units while increasing the amount of shade produced by those units. To establish which urban blocks in the city had the greatest available sunshine, parametric research was carried out. The relationship between shape, urban density, and the capacity for solar access using three different design criteria was investigated [45]. The criteria were as follows: (1) ground clearance, which is directly related to pedestrian thermal comfort; (2) access to daylight in the building facade, which indicates the function of daylight in the building; and (3) the use of solar cells on the building body, which indicates the amount of usable area in the building that can be used to install solar cells. This provides city planners and architects with some helpful options for coming up with concepts that can be applied in regions with dense textures (Figure 5).

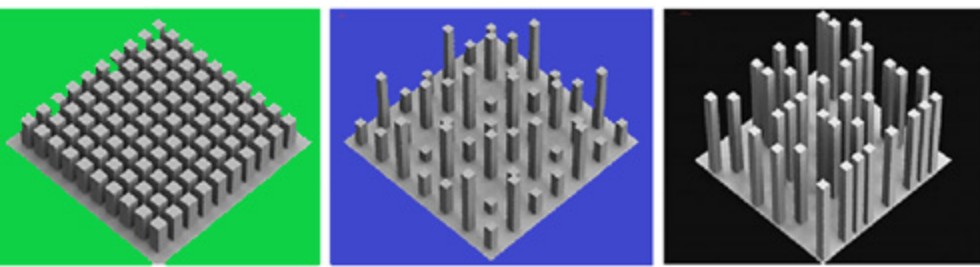

**Figure 5.** A variety of urban block distribution patterns with regard to the amount of sunlight in Baku.

Furthermore, it gives information on how to design an urban morphology for optimum sunshine access in a given urban density, where to locate the building to prevent shadows, and how to use street shape as a design parameter to improve the total energy performance of neighborhood units.

### 3.2. The Building's Facade Orientation

The orientation of the structure is yet another crucial component to consider when conducting a solar analysis of urban blocks. It determines the minimum amount of roof space necessary for solar heating and the installation of photovoltaic panels. The local conditions, the shape of the roof, and the orientation of the building all have an effect on the amount of solar energy that is absorbed [47]. Research was conducted on the geometric shapes of city blocks as well as how sun radiation is received by the different types of city blocks. An investigation into four urban blocks found that the layout and orientation of urban blocks had a significant impact on the amount of sunlight that could enter a building. Finally, it was concluded that access to sunlight was reduced by about 10–85 percent in urban blocks surrounded by other buildings (Figure 6).

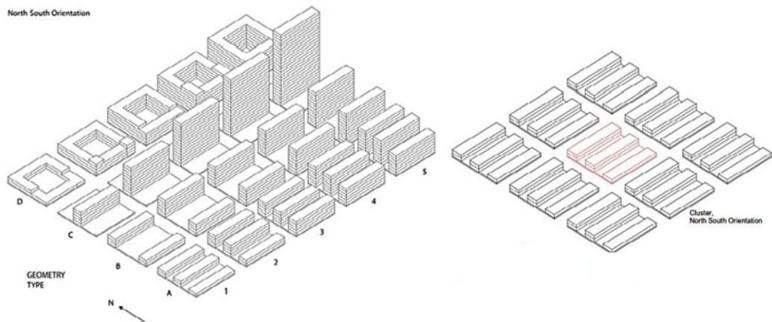

**Figure 6.** The component of the building's skin that is subject to the effects of solar orientation in Baku.

### 3.3. Design of Buildings and Streets

Many solar studies in a building are related to the structure's height to street width ratio. The shape of the building refers to its total volume, which should not exceed its borders. The vertical elevation of the building to the horizontal width of the roadway defines this ratio [48]. There have been numerous investigations conducted to determine the relationship between this ratio and the amount of solar energy received. In terms of shade, a larger H/W test along an axis reduces solar energy absorption in the vertical and horizontal surfaces of the building (Figure 7), while urban design parameters (street width and orientation) and building design parameters (roof shape and building shell design) affect the extent of access to sunlight in urban contexts, as well as the use of solar heating in residential buildings [49]. As a result, street width has the greatest impact on sunlight access, while street orientation has the least. Furthermore, access to solar energy has been compared in a variety of ways. In hot and dry climates, criteria such as building height, width, and street orientation are used to make this comparison.

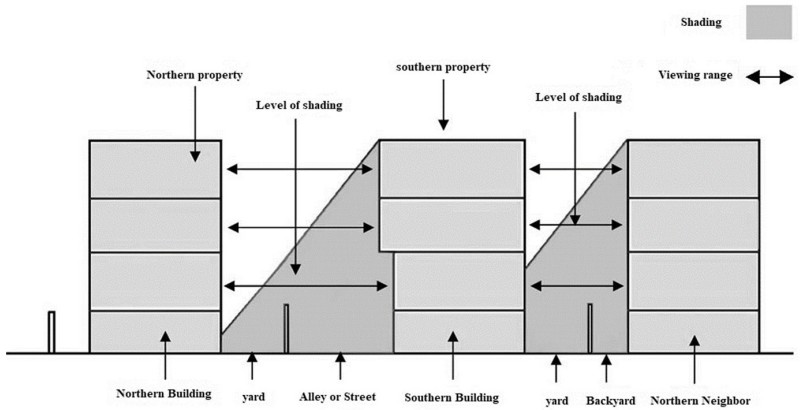

**Figure 7.** The relationship between the width of the street and the height of a building in order to maximize the amount of solar energy collected by the building in Baku.

### 3.4. The Effect of Shape Factors and Yard Orientation on the Thermal Behavior of the Building Was Investigated

In the climate of Baku, the location of the building mass inside the site is explored in six different general cases. The occupancy level, height, proportions, materials, and orientation of the structure are all the same across all models, and the building takes up sixty percent of the site. The dimensions of the land are 38 by 15, which, according to the approved occupancy level (which covers 65 percent of the property), is equivalent to 15 by 24. The building of it is not impossible. In order to determine the types of shapes that can be built on the ground, two close sites are combined and handled as if they were a single site. According to the ordinance of the municipality, a set may not build a window on the wall of an adjacent yard in any way. As a consequence of this, skylights should be installed prior to the creation of windows, and the windows themselves should be positioned within the skylights. The builder is granted permission to construct up to 70 percent of the length of the land, plus three meters, and must also include a skylight in the portion of the building that extends beyond the original footprint. This point emerges as an observation from the simulation of the model. Only the three bricks in the middle row of the construction block, which are arranged opposite each other, are analyzed for energy use. Each row of the construction block is evaluated for its seven individual bricks. The first case, dubbed "common," involves a common model with a manufacturing license in Baku. The building is positioned in the northern half of the site in this model. Skylights are provided in the surrounding wall of the southern blocks to provide light to the blocks. The linear model refers to the state in which the building is located in the far north and far south. The majority of the earth falls; in this situation, no patio is required, and the light of the blocks is delivered by the windows that are open to the block yard or street.

Each L-shaped block in the central courtyard structures is totally introverted. The section the L-shapes are turned in creates an open space between two nearby locations. Rotating the L-shaped blocks results in U-shaped urban layouts, both of which require a patio for optimal illumination. The total area of infrastructure created in all forms is equal to 600 square meters per floor and has four floors on the pilot. Six analyzed building shapes in three directions (facing south, turning 35 degrees to the west, and turning 35 degrees to the west) are simulated and the results are compared to evaluate the effect of building orientation on the consumption of cooling and heating energy inside the building (Table 1).

**Table 1.** Orientation in various shapes.

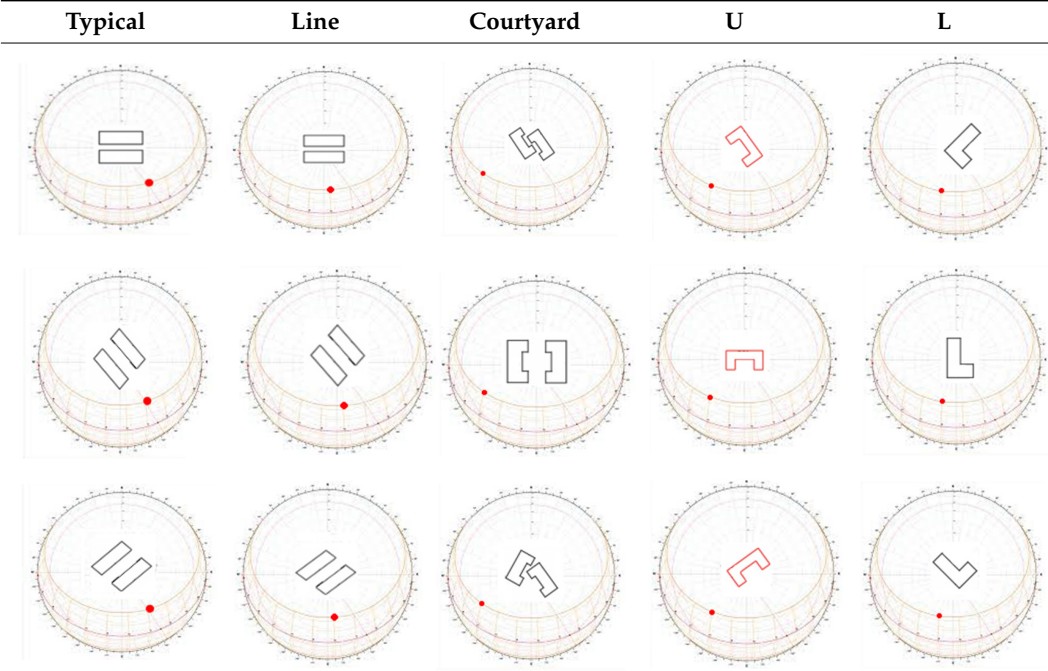

| Typical | Line | Courtyard | U | L |
|---------|------|-----------|---|---|

Figure 8 illustrates a number of different building types organized in a number of different sun angles, illustrating that L-shaped structures gain a significant quantity of light energy in comparison to other building layouts. This is demonstrated by the fact that buildings in the shape of an L gain the lightest energy of any other shape. On the other hand, the amount of sunlight that is received by entire building shapes is rather stable during the course of the year, while line-shaped structures only collect a minimum portion of the sun's heating radiation due to their orientation.

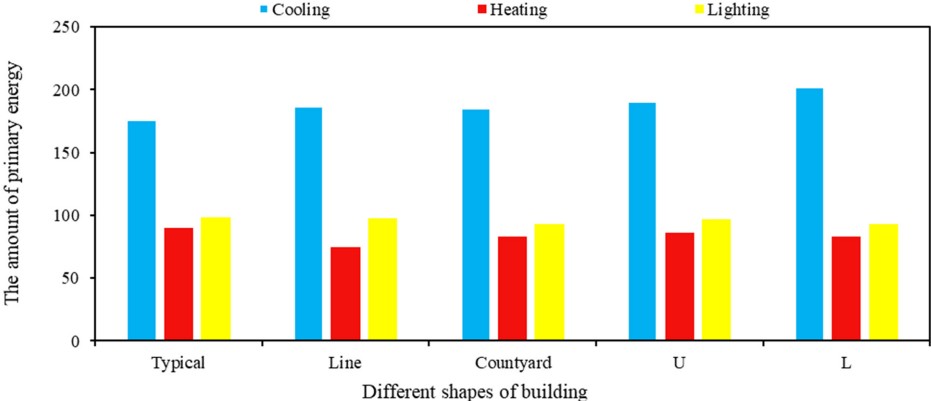

**Figure 8.** The amount of primary heating energy of different patterns per kilowatt hour per square meter per year.

As shown in Figure 9, the difference in the orientation of the building has the largest effect on the amount of energy consumption in both common and linear shapes; the highest difference, which reached 5 kWh/m$^2$, was between the rotation to the west and the orientation to the south. Being in the east and west directions has the lowest energy consumption compared to the other two modes in the three common forms, linear and extroverted, and there is not much difference in the energy consumption of different rotations in the other forms. Rotation to the west increase's energy consumption in all forms. In all forms, this species appears to be optimally orientated to the south and has the lowest energy and gas consumption.

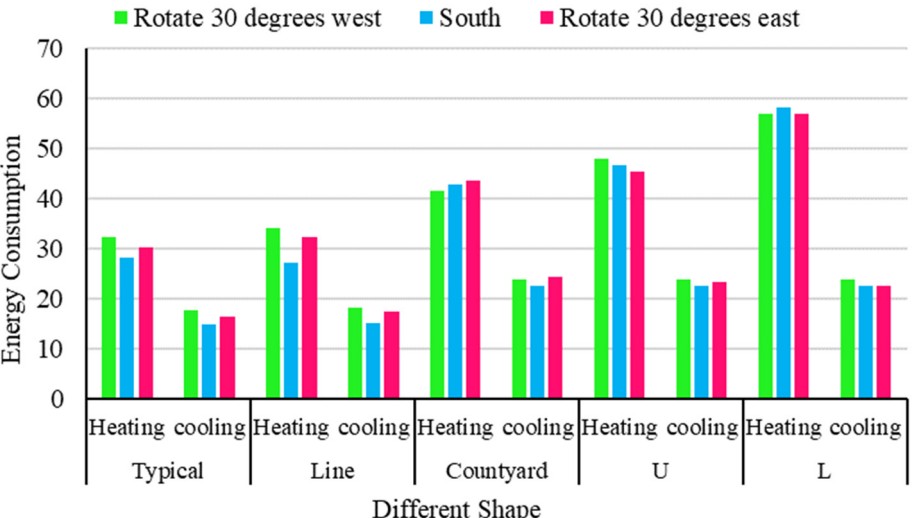

**Figure 9.** The application of energy for the purposes of heating and cooling in a wide variety of configurations.

When determining how much energy is needed for cooling in the summer, the amount of solar energy that can be absorbed is the most important factor to consider. As can be seen, the amount of solar energy absorbed by the building grows as it rotates to the east and west, with the exception of the L-shaped building, and this pattern is reflected in the amount of cooling energy that is used during the summer months. Only the L-shape, which faces east, reduces the quantity of solar radiation it is exposed to and, as a result, reduces the amount of cooling energy it needs to maintain the same temperature. In terms of ventilation, the building is more in the flow of cold air when the rotation is to the west, and when the rotation is to the east, the air flow increases, and the consumption of heating energy increases proportionately. In other words, there is not much of a difference in the quantity of heating energy that is used in the various directions, and the relationship between the charts that show energy consumption and the charts that show ventilation and radiation can be seen quite clearly. The difference in annual total energy use between heating and cooling is illustrated in Figure 10. It has been demonstrated that there is very little variation in the amount of energy consumed by buildings that rotate, and the only two common linear designs that spend less energy are those that stretch from east to west and fall between 4 and 6 kWh/m$^2$. The increase in the building's energy consumption, which can be seen in a few different forms, is the most important feature, and it is caused by the building's rotation to the west. As a direct consequence of this, any kind of rotation toward the west requires a greater amount of energy. It would appear that the orientation running north to south reduces the need for any and all sources of cooling energy during the summer.

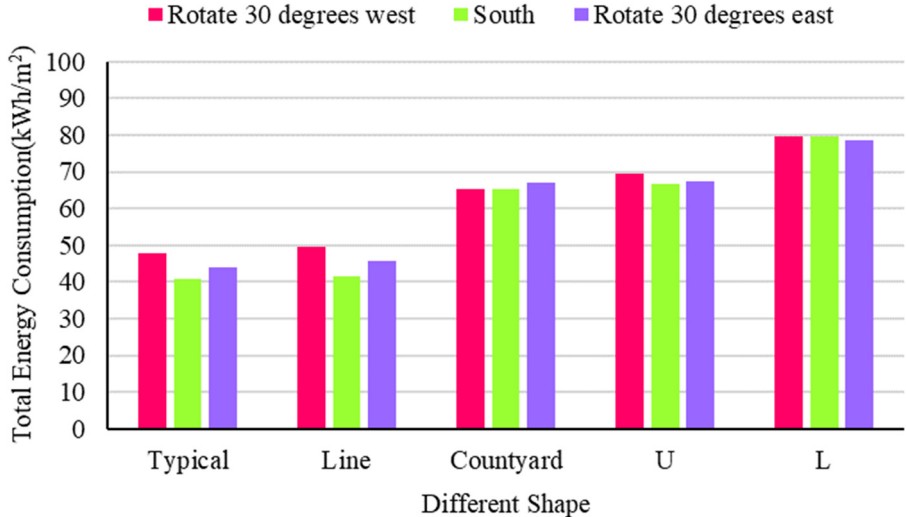

**Figure 10.** The comparison investigates overall energy use from a wide range of vantage points and organizational frameworks.

Analysis of variance and correlation were used to analyze the link between morphological indicators and primary energy use, and correlation coefficients were interpreted using Table 2. The following outcomes were obtained from the basic energy consumption simulation and correlation analysis: the correlation analysis results show a significant difference between building form, building height, mass location, block form, layout plan, and level of passages and open spaces, and a strong or high correlation between the amount of initial energy, while the index of block proportions has a moderate correlation with initial energy (Table 3).

**Table 2.** The amount of correlation coefficient.

| The Correlation Coefficient | Interpretation |
| :---: | :---: |
| 0–0.2 | Very low |
| 0.2–0.4 | Low |
| 0.4–0.7 | Middle |
| 0.7–1 | High |

**Table 3.** Correlation coefficient analysis.

| Independent Variable | The Dependent Variable | The Correlation Coefficient | Interpretation of the Correlation Coefficient |
| :---: | :---: | :---: | :---: |
| Shape of building | Primary energy | 0.856 | High |
| Building height | Primary energy | 0.712 | High |
| Shape of block | Primary energy | 0.689 | High |
| Block fit | Primary energy | 0.238 | Middle |
| Layout plan | Primary energy | 0.991 | Middle |
| The surface of roads and open spaces | Primary energy | 0.684 | High |

## 4. Conclusions

We made an effort to conduct literature research on the key factors that influence the amount of energy a building uses. Findings have demonstrated that it is particularly successful in the early stages of building design to recognize the effect of Fermi parameters on the passive solar design of buildings in order to maximize energy. Here at MATLAB, we make use of these studies to inform our work. When evaluating and comparing the

influence of each parameter, researchers used a variety of energy calculation software, and each researcher developed their own approach to the analysis. As a consequence of this, the objective was to determine the effect of each parameter on energy consumption rather than the calculating method and software that were utilized in the literature review. This review included research that evaluated and assessed the part that urban blocks play in the energy performance of buildings. The thermal behavior of a building is analyzed in each and every one of the studies, not as a single structure, but rather as a collection of individual buildings. In order to accomplish this, each study was classified according to one of these three categories:

- Investigates the effect that the shape of an urban block has on the thermal behavior of the building both inside and outside.
- Places emphasis on having access to sunlight both within and outside the structure so that static heating and lighting can be utilized.
- Examines the impact that the design of the urban block has on the ventilation of the building's interior as well as its exterior.

After reviewing the existing research on this subject, it was determined that it is extremely difficult to analyze the effect of neighboring units on the thermal behavior of the building since it is difficult to study all of the associated parameters at the same time. Furthermore, the methodologies and procedures available to assess all of these factors are extremely limited. Last but not least, due to the fact that solar radiation is the single most important aspect to take into account when analyzing the thermal behavior of a building, a significant amount of research has been conducted in this field. Furthermore, new methods for evaluating solar potential on an urban scale are currently being developed.

This analysis of the relevant research indicates that there are not a lot of studies that investigate the impact of urban block geometry on the flow pattern of wind outside while also investigating the flow pattern of air within the structure.

The form of the region's mantle is the most essential determinant in the creation of solar radiation energy in that location. The elements that form big local centers are the difference in height, the orientation of the plane (slope and direction), and the obstacles that prevent a point from accessing the energy. The radiant energy of the sun is also measured. Other factors, such as atmospheric transit and the relative position of the sun, affect this in addition to geography and the geometry of the globe.

Multiple neighborhood-scale morphological characteristics affect building energy use. Morphology and energy use are interconnected. Few physical metrics quantify urban tissue energy usage. Our study contrasted cooling, heating, and lighting and analyzed urban planning indicators. The research found many one-sided structures. Baku's northeast-southwest row blocks, average open space (50), and strip layout use less primary energy. Previous research showed that the square building form is acceptable for hot and dry conditions, but this model is optimal due to its centralized arrangement plan, mass placement in the middle of the plot, and vast distances between masses. Daylighting, heating, and cooling cannot dictate energy-efficient urban fabric patterns. All three factors must be balanced. Primary energy is affected by building height, mass placement, block form, layout design, and corridor and open space levels. Early in design, the morphological aspects' energy impact should be analyzed. Contemporary urban planning, the positive and negative repercussions of urban morphological features, and the correct judgment about urban layout in relation to energy use have been reviewed. Azerbaijan has long-standing energy and environmental policies. This research can be utilized to create urban development standards without preliminary research.

**Author Contributions:** Conceptualization, N.K.A.D. and N.B.K.; methodology, N.K.A.D.; software, U.R.; validation, I.P., N.B.K., and M.M.A.Z.; formal analysis, Y.F.; investigation, J.W.G.G.; resources, S.E.I.; data curation, T.A.; writing—original draft preparation, N.K.A.D.; writing—review and editing, U.R.; visualization, M.M.A.Z.; supervision, N.K.A.D.; project administration, N.B.K. All authors have read and agreed to the published version of the manuscript.

**Funding:** This research received no external funding.

**Data Availability Statement:** Not applicable.

**Conflicts of Interest:** The authors declare no conflict of interest.

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
