# Peer review of "An Analysis of Urban Block Initiatives Influencing Energy Consumption and Solar Energy Absorption"

_sustainability, doi:10.3390/su142114273_

Round 1

Reviewer 1 Report

This manuscript attempted to address the energy consumption of the building from the perspective of urban block and building scales. However, the title, the manuscript structure and its content are confusing.

Is this a "review" paper? You never mentioned it until "Conclusions"? In the "Material and Methods" section, you only highlighted and summarised other people's works, not yours. Figures 2 to 7, I assume, are from others too, right? And I can see that you cited up to [47], but your references only up to 39. And you always mentioned "he", "they", etc...who are they? Many incomplete and confusing sentences too. For example, in Line 202, "This study's findings are presented Gives urban planners and architects some beneficial solutions for designing in dense textures", and in Line 311, "Were discovered". In addition, if this (in section 3.4) is your work, how do you measure the energy consumption? If it is a simulation, what kind of software do you use?

They are 9 authors, and "All authors have read and agreed to the published version of the manuscript", without realising so many mistakes in the manuscript....

Author Response

Respond was attached

Reviewer 2 Report

This research paper addresses an interesting research topic regarding to an assessment of urban block initiatives affecting energy use and solar energy absorption, the paper needs to be reorganized and improved according to the following observations:
1. Please delete the abbreviations from the abstract
2. all abbreviations must be defined for the first time
4. All equations mentioned in the paper need an appropriate reference.
5. Conclusion section was weakly formulated. It should be improved and the most important results should be added and it will be clearer if the authors use numbered listing of the conclusions.
6. The current literature review is not sufficient. For example, please mention (give) following more current studies related to Solar energy, in the sections of Introduction, and References List for completeness of your study and the references:

·       Energy, exergy and exergy-economic analysis of a new multigeneration system based on double-flash geothermal power plant and solar power tower, Journal: Sustainable Energy Technologies and Assessments, Vol: 47

·       Exergoeconomic and environmental analysis of a combined power and water desalination plant with parabolic solar collector; Journal: Desalin. Water Treat, Vol: 193

7. Figure quality is poor. Some images are blur
8. Language ability is weak with various errors.
9. Figure 9 is too hard to understand
Authors should consider above mentioned remarks in order to revise the manuscript. Reviewer thinks that a publication of the draft manuscript may be possible after a major revision.

Author Response

Respond was attached

Reviewer 3 Report

Although the paper is potentially interesting, I think it should be improved in order to become potentially publishable to sustainability journal. In the current form, it is not publishable for the following reasons.

1.       The title “An assessment of Urban Block Initiatives Affecting Energy Use and Solar Energy Absorption” does not represent in a proper way the contents of the paper. In fact, if the authors talk about “An assessment.. “ they should show some methodological contribution, mathematical formulas, or empirical analysis. None of them are explained in a proper way in the paper. For these reasons the authors should use “An analysis…” rather than “An assessment…”.

2.       The abstract should be improved in order to give a clear idea about what is the goal of the paper and what is its contribution.  In the current form it is not very clear.

3.       Well done with the introduction and references. However the authors should add in the final part of this section the contents of the paper. For example, they could use the current form: “The paper is organized as follows: Section 2 describes….; Section 3…; etc”.

4.       Materials and Methods Section is too simple and I don’t see any contribution. I suggest to improve this section by describing in a more proper way the methodology with the mathematical models used by the authors. In the current form, the methodology is in its early stage and it must be improved.

5.       In general “Results and Discussion” are done following a data analysis with a strong empirical model (if the author contributes empirically to the existing literature), or by applying an innovative mathematical framework (if the author contributes methodologically to the existing literature). None of these cases is referred to the paper.

The  paper appears just a review of the literature  and so I suggest the rejection in the current form. To sum up, the authors should improve the contribution of the paper and they should explained the positioning of the paper in the previous literature.

I suggest to improve the paper following the line of the current title that considers as assessment of urban block initiatives. For example, the authors could provide a valuation methodology to price these projects.

Author Response

Respond was attached

Reviewer 4 Report

The authors have chosen a title that is really worthwhile. I believe the manuscript could be accepted for publication in this journal with minor revisions and after address the following comments.
1.    In the abstract, methodology and models used in the study should clarify by new sentences.
2.    The last paragraph of the introduction section (Line 121 to 130) and novelty of the manuscript should explain more and clear this style was a bit unclear.
3.    The method used in the research was so vague, that I recommend adding a flowchart into material and method section of the manuscript for more clarification.
4.    The figures well done but caption should revised and give more information for readers.
5.    The use of a spell checker is obligatory in the English language and style.
6.    Some references in the text are not available at the end of the manuscript e.g. [40] to [48].

Author Response

Respond was attached

Reviewer 5 Report

Following comments recommended improving the manuscript (Minor revision): 1. The novelty of the study's originality needs to be mentioned in the abstract. This version does not make it abundantly apparent. 2. More quantitative results should be mentioned in the abstract than qualitative results. 3. The Author could use a multi objective optimization methods to achieve best orientation for blocks. 4. Discussion about results was so simple. It needs to revise extensively. It was not clear the objective of the manuscript.  5. Line 284 to 300 “The amount of solar energy absorption is the criterion for analyzing the amount of energy consumed for cooling in the summer. Except for the L-shape, the absorption of so-lar energy increases with the rotation of the building to the east and west, as can be seen, and this pattern is reflected in the use of cooling energy in summer. By turning eastward, only the L-shape receives less solar radiation and consumes less cooling energy in the same amount. In relation to ventilation, the building is more in the flow of cold air when the rotation is to the west, and when the rotation is to the east, the air flow increases and the consumption of heating energy increases proportionately. In other words, the amount of heating energy consumed in various directions is not much different, and the relation-ship between energy consumption charts and ventilation and radiation charts can be clearly seen. Figure 10 depicts the annual difference in total energy use for heating and cooling. As can be shown, there is little change in energy consumption with building rota-tion in most forms, and only two frequent linear designs between 4 and 6 kWh/m2 east-west stretch spend less energy. The crucial aspect is the rise in energy consumption caused by the building's rotation to the west, which can be seen in several forms. As a re-sult, all sorts of rotation to the west consume more energy. The north-south orientation appears” This paragraph is unclear. I am not sure what do you want to say. 6. Conclusion section write so short. It has to be expressed in a more detailed manner, with the primary emphasis being placed on the primary findings of the study.

Author Response

Respond was attached

Reviewer 6 Report

I reviewed the article. I decided it was publishable.

Author Response

Thank you so much for reviewing the manuscript.

Round 2

Reviewer 1 Report

This manuscript is the second attempt to address the energy consumption of the building from the perspective of urban blocks and building scales. Although it is much improved, still, some previous comments still go unanswered. For example, "This study's findings are presented Gives urban planners and architects some beneficial solutions for designing in dense textures (Figure 5)" in line 202 (original manuscript) or in line 237 (revised manuscript). It is unclear what the authors were trying to say. Please check all sentences for any unclear or grammatical errors.

Author Response

With thanks for your comment, we did alter and increase the literature.

Reviewer 3 Report

The paper has been greatly improved. The methodology has been improved in terms of calculations, and so the work appears more publishable in this form. However, The English should be improved. For example, pag 4 line 155 :"In Section 2 introduces..". This is not correct. The authors should use "Section 2 introduces" or "In Section 2 we introduces..".

So let me suggest minor revisions of the work.

Author Response

Thank you for your comments.
